# Design of Rational JAK3 Inhibitors Based on the Parent Core Structure of 1,7-Dihydro-Dipyrrolo [2,3-b:3′,2′-e] Pyridine

**DOI:** 10.3390/ijms23105437

**Published:** 2022-05-13

**Authors:** Yihao Li, Dan Meng, Jiali Xie, Ruoyu Li, Zifan Wang, Jinlong Li, Lin Mou, Xinhao Deng, Ping Deng

**Affiliations:** 1College of Pharmacy, Chongqing Medical University, Chongqing 400016, China; lyh1620885216@hotmail.com (Y.L.); mengdan429@stu.cqmu.edu.cn (D.M.); xiejiali@stu.cqmu.edu.cn (J.X.); lry@stu.cqmu.edu.cn (R.L.); w1347191500@163.com (Z.W.); ljl18388503873@outlook.com (J.L.); ml249796467@outlook.com (L.M.); d15820985768@outlook.com (X.D.); 2Chongqing Research Center for Pharmaceutical Engineering, Chongqing 400016, China; 3Chongqing Key Research Laboratory for Quality Evaluation and Safety Research of APIs, Chongqing 400016, China

**Keywords:** JAK3, parent structure design, backbone growth, molecular docking, molecular dynamics

## Abstract

JAK3 differs from other JAK family members in terms of tissue distribution and functional properties, making it a promising target for autoimmune disease treatment. However, due to the high homology of these family members, targeting JAK3 selectively is difficult. As a result, exploiting small changes or selectively boosting affinity within the ATP binding region to produce new tailored inhibitors of JAK3 is extremely beneficial. PubChem CID 137321159 was used as the lead inhibitor in this study to preserve the characteristic structure and to collocate it with the redesigned new parent core structure, from which a series of 1,7-dihydro-dipyrrolo [2,3-b:3′,2′-e] pyridine derivatives were obtained using the backbone growth method. From the proposed compounds, 14 inhibitors of JAK3 were found based on the docking scoring evaluation. The RMSD and MM/PBSA methods of molecular dynamics simulations were also used to confirm the stable nature of this series of complex systems, and the weak protein–ligand interactions during the dynamics were graphically evaluated and further investigated. The results demonstrated that the new parent core structure fully occupied the hydrophobic cavity, enhanced the interactions of residues LEU828, VAL836, LYS855, GLU903, LEU905 and LEU956, and maintained the structural stability. Apart from this, the results of the analysis show that the binding efficiency of the designed inhibitors of JAK3 is mainly achieved by electrostatic and VDW interactions and the order of the binding free energy with JAK3 is: 8 (−70.286 kJ/mol) > 11 (−64.523 kJ/mol) > 6 (−51.225 kJ/mol) > 17 (−42.822 kJ/mol) > 10 (−40.975 kJ/mol) > 19 (−39.754 kJ/mol). This study may provide a valuable reference for the discovery of novel JAK3 inhibitors for those patients with immune diseases.

## 1. Introduction

Janus kinase (JAK) plays a central role in the regulation of the immune system and has become an important drug target for the treatment of many immune diseases [1]. The JAK kinase family consists of four members in humans: JAK1, JAK2, JAK3, and TYK-2, each of which has about 1100 amino acids and can be classified into seven homologous structural domains in order from the C-terminus to the N-terminus [2]. Each of them can bind to different cytokines to facilitate signaling to interleukins (IL), interferons (IFN), and other cytokines. These cytokines include IL-2, IL-4, IL-7, IL-9, IL-15, and IL-21 [3]. However, of these four isoforms, only JAK3 is predominantly expressed in hematopoietic tissue cells and is uniquely associated with cytokines shared by common γ-chain receptor subunits [4]. JAK3 binds to the γ_c_ receptor in the cytoplasm and works in concert with JAK1 to activate/phosphorylate signal transducer and activator of transcription (STAT) proteins. When phosphorylated, these STAT proteins dimerize and translocate to the core to initiate gene transcription, thereby specifically activating T, B, and NK cell immune responses [5]. Therefore, selective targeting of JAK3 has attracted many interests to prevent transplant rejection, treat various autoimmune diseases such as rheumatoid arthritis [6], ankylosing spondylitis (AS) [7], and psoriasis [8], and reduce the adverse effects of JAK1 and JAK2 inhibition [1].

However, traditional ATP-competitive inhibitors have been difficult to achieve high selectivity within the JAK family because all JAK family members have substantially similar enzyme activity sites [9]. Fortunately, one of the few fundamental structural differences in JAK3 is the discovery of a cysteine residue (CYS909) located in the solvent-exposed front of the ATP activity pocket [10,11]. This amino-acid residue is substituted by a serine in the other three isoforms of the JAK kinase family [12] which provides an opportunity to achieve JAK3 targeting selection through covalent inhibition. To date, most selective JAK3 inhibitors reported in the literature have similar parent core structures (Figure 1). Thorarensen et al. reported PF-06651600, an inhibitor based on a 6,7-dihydro-5H-pyrrolo [2,3-d] pyrimidine structure with an acrylamide warhead that irreversibly binds to Cys909 in JAK3. However, the collective consequences of inhibiting JAK3 in a completely irreversible covalent manner remain unclear and are not beneficial for all purposes [13]. London et al. sidestepped this problem by designing reversible covalent JAK3 inhibitors using cyanoacrylate warheads [14]. Forster et al. built on this to design reversible covalent JAK3 inhibitors with a tricyclic structure that exhibited both high isomeric and dynamic group selectivity and strong cellular activity and selectivity [15]. In addition, structural optimization of AZD1480 by AstraZeneca revealed that filling the hydrophobic pocket around the hinge region significantly improved the inhibition of JAK2 by the compound [16], and thus this principle could be applied to selective JAK3 inhibitors. Unfortunately, however, none of the JAK3 parent nucleus structures reported in the literature can fully utilize the solvent-exposed region of the ATP-active pocket and fill the hydrophobic pocket. To this end, we have used PubChem CID 137321159 as the characteristic part of the lead inhibitor for assembling and redesigning the new parent core structure to fill the hydrophobic pocket as well as to deliver reversible covalent warheads into the expected sites. 

## 2. Results and Discussion

### 2.1. Structural Design of JAK3 Inhibitors

The high similarity of the ATP-binding sites in the JAK kinase family poses a considerable challenge for the development of isoform-selective inhibitors. The unique cysteine (CYS909) in the solvent-exposed region of the JAK3 active pocket corresponds to the serine in other JAK kinase family members, providing an opportunity to achieve JAK3 isoform selectivity through covalent targeting. However, a considerable challenge in using covalent kinase inhibitors (CKI) lies in their potential to react with other kinase targets that contain cysteines at the same relative position. Interestingly, it was found in the binding models of the ligand-target crystal structures that the introduction of moieties or individual atoms for filling in hydrophobic or lipophilic active pockets that are not filled or are inadequately filled may result in a substantial increase in affinity and enhanced activity, which suggests that filling small voids within the embedding cavity is important for drug design. Based on the above, in this study, two means of replacing the reversible covalent warhead and redesigning the parent core structure were attempted to improve the selectivity of the JAK3 inhibitor.

The active pocket of JAK3 kinase (Figure 2) mainly consists of a hydrophilic pocket at the front end (ARG911, ASP912 and AGR953) and a hydrophobic pocket at the back end (LEU828, VAL836, ALA853, LYS855, VAL884, MET902, GLU903, LEU905, LEU956, ALA966 and ASP967). Further observation of the kinase structure reveals GLU903 and LEU905 residues at the bottom of the hydrophobic pocket that can form hydrogen-bond interactions with the inhibitor. If inhibitors could fully occupy these active sites, they might yield higher affinity and selectivity.

The structure of inhibitor 5 was first optimized for the design of a three-ring parent-core structure, namely 1,7-dihydro-dipyrrolo [2,3-b:3′,2′-e] pyridine, and the cyanamide was replaced by the cyano-acrylamide, resulting in the novel inhibitor 6 (Figure 3). Inhibitor 6 consists of four components: the parent core, the linker, the reversible covalent warhead, and the R-group, which are marked here in red, magenta, blue, and light blue, respectively. Of inhibitor 6, the parent core is located in the hairpin structure of the hydrophobic region and forms hydrogen-bonding interactions with residues GLU903 and LEU905 to fill the pocket cavity and expel water molecules. Meanwhile, the warhead is placed in the solvent-exposed region of the active pocket to fully contact CYS909. Furthermore, the 1,2-hydrindene structure of the linker forms hydrophobic interactions with LEU828, VAL836 and LEU956, and the oversized active pocket in the middle of the hydrophobic region provides an opportunity for reversal of the linker. Therefore, to counteract the strong torsional force of the linker, an R-group was added to the 1,2-hydrindene of the linker, which may position more interactions between the inhibitor and the receptor residues LYS855 and ASP967. 

A total of 1468 inhibitors, including the parent core structure, were generated using the backbone growth method. The inhibitors were then batch-processed using the Pre-pare Ligands module and Minimize Ligands module in DS. Using the crystal structure (PDB ID:5LWN) as the receptor, molecular docking was performed using AMDock and the “CDOCKER” module of DS, respectively. Then, the obtained results were analyzed to ensure the correct binding model using DS software, which can also provide more sufficient information for the inhibitor–kinase interactions.

### 2.2. Analysis of Activity and Selectivity of the Inhibitors

To gain insight into the docking pose of inhibitors at the active site of the receptor and to verify the correct conformations of the inhibitors, molecular docking experiments were carried out. Firstly, the “CDOCKER” module of DS was used to retain the top 15% inhibitors in the “-CDOCKER ENERGY” score, and the inhibitors with unreasonable structures were manually deleted. Thus, a total of 191 inhibitors were obtained. Next, these 191 inhibitors were docked to the ATP-competitive binding site of JAK3 using AutoDock Vina software, and only 14 inhibitors with an affinity greater than inhibitor 4 (−9.8 KJ/mol) were retained. The binding poses of the 14 inhibitors to the crystal structure (PDB ID:5LWN) at the active site were visualized by DS software, and detailed information about the key amino-acid residues associated with hydrogen bonding interactions and tight contacts were examined (Table 1).

To verify whether the R-group meets the expectations of the structural design, the focus here was put on the binding mode of the R-group to the active site. Finally, only inhibitor 6, inhibitor 8, inhibitor 10, inhibitor 11, inhibitor 17, and inhibitor 19 counteract the linker reversal and can deliver reversible covalent warheads into the expected sites. Inhibitor 8 with the highest affinity—of which R-group penetrates deep into the active pocket and forms a pi-anion interaction and a hydrogen bond between the fluorotoluene fragment with ASP967 and LYS855, respectively—may be one of the reasons for counteracting the linker’s torsion. Meanwhile, the secondary amine group of the linker forms a hydrogen bonding interaction with AGR953, which works together with the R-group to counteract the torsion of the linker, which may send the olefin in cyanoacrylate to a position more dependent on CYS909 and facilitate a Michael addition with CYS909. A close-up look of Inhibitor 8 is shown in Figure 4.

The parts of this series of inhibitors other than the R-group have similar interactions at the active site. The binding pattern of the R group of partial inhibitors is depicted in Figure 5. As expected, the parent core of the tricyclic structure of all inhibitors appears not only in the center of the hydrophobic pocket but also in the region where the water molecule was originally located. Specifically, the alternating hydrogen-bond donor and acceptor formed by 1,7-dihydro-dipyrrolo [2,3-b:3′,2′-e] pyridine interact with GLU903 and LEU905 at the bottom of the hydrophobic pocket to form three crucial hydrogen bonds which anchoring the ring structure at the bottom of the active site. In parallel, the new parent core also forms hydrophobic interactions with LEU828, ALA853, and LEU956, which may be related to substrate recognition. In addition, since this series of compounds shares a similar reversible covalent warhead with inhibitor 4, the cyano-acrylamide is similarly located near CYS909, AGR911 and AGR953 (Figure 3 and Figure 4). Although the other inhibitors are slightly lower in affinity than inhibitor 8, they are similar to inhibitor 8 in terms of docking posture at the active site and are expected to enter the next experimental potential.

### 2.3. ADMET Analysis

The physicochemical properties of a drug are closely related to its pharmacokinetic properties and bioactive strength. Therefore, understanding and calculating the physicochemical properties of drugs is essential for drug development, and ADMET properties include several interrelated processes of absorption, distribution, metabolism, and excretion, which are related to the bioavailability of drugs in vivo.

In this research, SwissADME and ADMETlab 2.0 web servers were used to calculate the pharmacokinetic characteristics and toxicity of inhibitors in humans. The results showed that all inhibitors except inhibitor 10 and inhibitor 11 satisfied Lipinski Rule (Table 2). While none of the six inhibitors possessed cytochrome P450 CYP2D6 enzyme inhibitory activity. In addition, the blood–brain barrier permeability model showed that all inhibitors did not penetrate the blood–brain barrier, but all had excellent human intestinal absorption. For carcinogenicity prediction, all six inhibitors were predicted to be low in carcinogenicity. However, only three inhibitors had a moderate affinity for plasma proteins.

Despite the relatively poor pharmacokinetic characteristics of some of the inhibitors, some deviation in ADMET properties may be acceptable if the drug molecules exhibit the desired pharmacological properties.

### 2.4. MD Simulation Analysis

To examine the stability of the ligand–protein system in an aqueous solution, the best docking conformations of the six inhibitors which could counteract the linker reversal with the JAK3 were used as the initial structures for the molecular dynamics simulations. Molecular dynamics simulations of the docking complexes were used to validate the docking results and to analyze the dynamic motion of the docking complexes to understand their stability. 

The stability and convergence of these systems were determined by the RMSD of the ligand relative to the backbone Cα atom. 5LWN-6 and 5LWN-10 complex systems showed very similar RMSD trends, with both RMSD values rapidly increasing from 0 to 0.3 nm, and all systems reaching equilibrium in less than 5 ns of simulation time. During the subsequent simulations, the RMSDs of both the 5LWN-6 complex system and the 5LWN-10 complex system also remained around 0.3 nm. Meanwhile, the RMSD values of the 5LWN-8 complex system underwent an increase and then a decrease during the simulation time of 0–20 ns, with a rising maximum value of 0.5 nm; its value converged and stabilized around 0.4 nm after 20 ns of simulation, with the smallest fluctuation among all inhibitors (about 0.05 nm). In particular, the RMSD values of the 5LWN-8 complex system were slightly higher than those of the 5LWN-6 complex system, possibly because of a flexible R-group incorporated in the inhibitor 8, which gave the system a high RMSD. All of the above three complexes were able to reach the equilibrium state in a relatively short time although there were slight fluctuations during the simulations, these complexes did not fluctuate excessively, indicating that the model reached equilibrium, which may be related to the fact that all of the R-groups contain fluorobenzene structures (Figure 6). 

Unfortunately, the 5LWN-11, 5LWN-17 and 5LWN-19 complex systems showed a high level of RMSD fluctuation (about 0.1 nm) and unstable trajectory throughout the 100 ns simulation time. Except for 5LWN-17, no significant convergence was found in the final trajectories of the two complexes, reflecting the instability of their complex systems.

To further explore the binding mode of the tricyclic structure inhibitors to JAK3, differences and similarities were obtained by comparing the hydrogen bonding characteristics of the six inhibitors. There were also notable differences among these six complex systems. The three complex systems, 5LWN-8, 5LWN-10, and 5LWN-17, all stably maintained four hydrogen bonds during the simulation time, were immobilized at the bottom of the hydrophobic pocket. The number of hydrogen bonds in the 5LWN-11 complex system fluctuated around three to four, which may explain the certain degree of fluctuation in its RMSD value. However, the 5LWN-6 and 5LWN-19 complex systems exhibited a different hydrogen-bonding profile from the first four. Although they were able to maintain more than five hydrogen bonds at the beginning, the number of hydrogen bonds gradually decreased with increasing simulation time, and eventually fluctuated at the level of three to four hydrogen bonds. To understand the stability of the parent core structure in the active pocket, a total of five frames of protein–ligand conformations were extracted at 25,000 ps intervals and hydrogen-bonding interactions were observed (Appendix A). As expected, the number of hydrogen bonds of the protein–ligand in all five frames was larger than that of the original crystal complex system in the simulation time of 100 ns, and the fragment of the parent nucleus structure was stabilized at the bottom of the hydrophobic pocket to form three hydrogen bonds with residues GLU903 and LEU905, showing no signs of detachment from the active pocket. In contrast, the 5LWN-4 complex had only two hydrogen bonds, and the parent core tended to detach from JAK3 during the simulation. This demonstrates the superiority of the newly designed parent core.

### 2.5. Evaluation of the Binding Energy

To evaluate the binding affinity of the inhibitors, including a tricyclic structure within the ATP-competitive binding site of JAK3, the binding free energies of the inhibitors bound to JAK3 were calculated using the MM-PB/SA method using the g_mmpbsa program based on the docking analysis. Here, the binding free energy was composed of van der Waals energy, electrostatic energy, polar solvation energy, and SASA energy (Table 3). Notably, van der Waals energy and electrostatic energy were the main contributors to the corresponding total values due to the hydrophobic contacts of the inhibitors with the nonpolar residues of the hydrophobic pocket and the proximity of the cyanoacrylate to the hydrophilic pocket in the solvent-exposed region where CYS909 was located.

Calculations show that the order of Δ*G*_bind_ is inhibitor 8 (−70.286 kJ/mol) > inhibitor 11 (−64.523 kJ/mol) > inhibitor 6 (−51.225 kJ/mol) > inhibitor 17 (−42.822 kJ/mol) > inhibitor 10 (−40.975 KJ/mol) > inhibitor 19 (−39.754 kJ/mol). Among them, only the binding free energy of inhibitor 8 with JAK3 is higher than that of the 5LWN−4 complex system. In addition, Δ*E*_vdw_ contributions were found to have similar values (from −186.017 to −204.047 kJ/mol) in all six inhibitor systems, which is likely to involve the same parent core and linker structure in these molecules. However, Δ*E*_ele_ contributions could explain the difference in their binding. Specifically, the order of Δ*E*_ele_ contributions was inhibitor 10 > inhibitor 6 > inhibitor 19 > inhibitor 17 > inhibitor 8 > inhibitor 11, where the corresponding Δ*E*_ele_ values were −82.609 kJ/mol, −73.232 kJ/mol, −65.409 kJ/mol, −64.784 kJ/mol, −54.868 kJ/mol, and 23.361 kJ/mol, respectively. These data do not correlate positively with the number of hydrogen bonds, which is possibly due to the formation of other weak interactions of the R-group with residues in the active site.

Because of the similar structures of the six inhibitors, the contributions of the key amino acids within the active pocket to the binding free energies were also calculated in order to understand the details of the inhibitor−receptor interactions. The number of hydrogen bonds was increased by the existence of the parent core, the novel tricyclic structure, thus enhancing the contribution to Δ*E*_ele_ in comparison with inhibitor 4. In addition to the hydrogen-bonding interactions, additional conjugation interactions with LEU828, ALA853, and LEU956 were formed for the tricyclic structure, either. In particular, the tricyclic parent core enhanced the interaction with residue LEU828, of which contribute to Δ*E*_bind_ (average −6.261 kJ/mol) was much larger than that of 4 (−0.479 kJ/mol). Considering the above facts, this is further evidence of the superiority of the tricyclic parent core structure (Appendix A).

Although the binding free energy of inhibitor 8 with JAK3 is not a significant advantage compared to other inhibitors, the difference was reflected in the value of the contribution of key residues to the binding free energy. Specifically, the parent core of inhibitor 8 formed a conjugate interaction with LEU956, which made the greatest contribution with the corresponding highest value (−11.371 kJ/mol) among all complex systems (Appendix A). To explore the reason for this, it was likely to be due to the hydrogen-bonding interaction formed by the fluorotoluene fragment of the R-group with LYS855, which could stabilize the linker and thus make a great contribution for the ligand to LEU956. This can be corroborated by its value of Δ*E*_MM_ (−5.589 kJ/mol) (Appendix A).

In conclusion, inhibitor 8 has a high value of the binding free energy to JAK3, thus stabilizing the conformation of the protein, and the main reason was due to the existence of the R-group, which contributes to the van der Waals interaction on the nonpolar surface and hydrogen bonds at the active pocket.

### 2.6. Weak Interaction Analysis

Weak interaction analysis can further analyze the favorable and unfavorable interactions between the receptor and ligand, and also complement the hydrogen-bonding analysis, the spatial repulsion, and the van der Waals interaction. In this study, weak interaction analysis was used to describe the mechanism of JAK3−inhibitor interactions and was complemented by plotting aRDG plots colored by the thermal fluctuation index TFI (Figure 7 and Appendix A). Green is the predominant color on the equivalence surface between the inhibitor and JAK3, which indicates that the van der Waals interaction is the main factor in the binding efficiency of the inhibitor to JAK3. In the six systems, there was a small difference in the Δ*E*_vdW_ contribution due to the similar area of the green contour (except near the R-group of inhibitor 17 where there was almost no the van der Waals effect), which was consistent with the results of the binding free energy calculations. In addition, several segments of cyan contours were found near the R-group of inhibitor 10, which could explain its great Δ*E*_ele_ bias.

## 3. Materials and Methods

### 3.1. Receptor Preparation

So far, a large number of X-ray crystal structures of JAK3 have been reported in the PDB database (http://www.rcsb.org, accessed on 12 December 2021) with various inhibitors, including 3LXL [19], 4HVD [20], 4QPS [21], 5LWN [15], 5LWM [15], 5TTU [13], 6DB4 [22], 6DUD [22], and 6GL9 [23]. Because the inhibitor 79R (inhibitor 4) in 5LWN has a specific structure of a reversible covalent warhead and the best resolution of the crystal structure, the protein receptor from 5LWN was chosen to dock all inhibitors after the protein preparation process. The receptor protein (PDB ID: 5LWN) was then prepared using the Discovery Studio 2020 software package (DS: https://www.3ds.com/, accessed on 12 December 2021) downloaded from the PDB database using the Prepare Protein module. The following tasks were performed: construction of the missing loop region, optimization of the side-chain conformation, removal of cocrystallized water molecules, and addition of hydrogen atoms.

### 3.2. Scaffold Growth and Ligand Preparation

The Grow Scaffold module in the Discovery Studio 2020 software package is used to select an atom or a group in the parent molecule as a reaction site and select classical chemical reactions for fragment growth, e.g., amide synthesis reactions, ether synthesis reactions, etc. The MM−GBMV/SW model can be selectively used to optimize the ligand or selected part of the side chain of the protein. Finally, the inhibitors are enumerated and Pareto optimized according to the matching properties within the protein pocket. Therefore, the molecules designed by this method are easier to synthesize experimentally (Figure 8).

The generated inhibitors were processed by using the Prepare Ligands module in the Discovery Studio 2020 software package to generate three-dimensional structures and assign charges to the inhibitors. Up to 10 stereo conformations and low-energy conformations were preserved per inhibitor. Finally, structural optimizations of the inhibitors based on the CHARMm force field were performed using the Minimize Ligands module.

### 3.3. ADMET Prediction

ADMET represents the five main features of pharmacokinetics: absorption, distribution, metabolism, excretion, and drug toxicity [24]. ADMET screening was performed to obtain pharmacodynamically compatible inhibitors to reduce the loss of drug discovery and development. Pharmacokinetic properties and toxicity of drugs in humans were assessed by Swissadme [25] and ADMETlab 2.0 [26] web server, including human intestinal absorption (HIA) plasma protein binding (PPB), blood−brain barrier (BBB), cytochrome P450 CYP2D6 binding, hERG Blockers, and carcinogenicity.

### 3.4. Molecular Docking

The “CDOCKER” [27] and Autodock vina [28] programs were both performed for the molecular docking studies because they have been widely used for docking to discover potential drugs with different targets. The Docking Ligands (“CDOCKER”) protocol, a lattice-based molecular docking method, uses the CHARMm force field in the Discovery Studio 2020 software package. In the study, the receptor-binding site was positioned at the area where the ligand molecule 79R (inhibitor 4) was located, as determined in the PDB site record. “Top Hits” was set to “20”, “Pose Cluster Radius” was set to “0.5 “, and other settings were default values. Only one optimal docking pose was recorded for each molecule and saved for further analysis.

Autodock vina is the molecular docking developed by Oleg et al. [28]. Molecular docking studies were performed with the Autodock vina module in AMDock 1.5.2 [29] to analyze the binding mode of designed inhibitors at the active site. The number of docked conformations was set to 20 poses in AMDock, and a box was placed in the geometric center of the existing ligand by selecting “Center on Hetero”, with the chosen center setting and GB size (x = −26.28, y = 12.60 and z = 58.96) and (x = 52, y = 68 and z = 68), respectively. Simple docking” was chosen to predict the binding pattern of individual protein−ligand complexes. After docking, the binding affinities of the different binding poses were ranked from high to low, while the docking conformations of each molecule with better docking scores were retained.

### 3.5. Molecular Dynamic Simulation

Molecular dynamics (MD) of the protein−ligand complexes was performed by selecting the best binding mode at the active site of 5LWN according to the docking scoring and the correct binding mode compared to known inhibitors in the corresponding crystal structures. MD simulations were performed using GROMACS 2020.3 software [30]. The required files, such as molecular topology files, molecular structure files, kinetic parameter files, run input files, and trajectory files, were obtained through GROMACS. The bond order, protonation, and reciprocal isomerization states of the ligands were checked using Avogadro [31] and the CGenFF [32] web server (https://cgenff.umaryland.edu/, accessed on 10 January 2022) was used to perform the atomic typing as well as the analogous assignment of parameters and charges and to generate the topological parameters of the ligands in a fully automated manner. Protein topologies were prepared using the CHARMm36 force field [33]. MD simulations were performed in explicit solvents under periodic boundary conditions, and each complex was immersed in a cube of 1*1*1 nm size filled single-point-charge (SPC) water molecules. The systems were stabilized by the addition of sodium salt/chlorine ions. Before the MD run, the energy of each system was minimized by using the steepest descent integrator in 5000 steps with a maximum force below 1000 kJ/mol or no drastic energy change. Afterward, NVT and NPT simulations were performed separately to equilibrate the systems [34]. V-rescale and Parrinello−Rahman were used to equilibrate the system, respectively, where the total pressure and temperature were 1 bar and 310 K, respectively, for 100 ps to reach a steady state. The linear constraint solver (LINCS) algorithm was used to solve for the constrained bond lengths [35]. Finally, the model performed a 100 ns MD simulation with trajectories recorded at 20 ps intervals. After the MD simulation was completed, the root-mean-square deviation (RMSD), and the number of hydrogen bonds was estimated to check the conformational changes and stability of the protein−ligand complexes. 

### 3.6. MM−PBSA Free Energy Calculations

The binding free energies of the receptor−ligand complexes were estimated by the g_mmpbsa [36] tool based on the molecular mechanics Poisson−Boltzmann surface area algorithm. For each complex, 1001 snapshot structures were extracted at 20 ps intervals from the last 20 ns along the MD trajectory and then used to calculate the binding free energy. The major residues that played an important role in the binding of free energy provide clear insight into the molecular mechanism of protein−ligand interactions. In this method, the binding free energies were calculated using the following equations [37]:Δ*G*_bind_ = *G*_complex_ − (*G*_receptor_ + *G*_ligand_)(1)
Δ*G*_bind_ = Δ*E*_MM_ + Δ*G*_sol_ − *T*Δ*S*(2)
Δ*E*_MM_ = Δ*E*_ele_ + Δ*E*_vdw_(3)
Δ*G*_sol_ = Δ*G*_PB_ + Δ*G*_np_(4)
Δ*G*_np_ = γΔ*SASA*(5)
where Δ*G*_bind_ is the free energy of binding; Δ*E*_MM_ is the difference in the internal energy of molecules under vacuum; Δ*G*_sol_ is the difference in the free energy of solvation; *T* is the thermodynamic temperature, and Δ*S* is the entropic change. Δ*E*_MM_ includes the electrostatic interaction called under vacuum and the van der Waals interaction under vacuum. Δ*G*_sol_ consists of the difference in the free energy of polar solvation (Δ*G*_PB_), the difference in the free energy of nonpolar solvation (Δ*G*_np_), and the polar part (Δ*G*_PB_) is obtained by solving the finite-difference Poisson−Boltzmann equation. The nonpolar part (Δ*G*_np_) is obtained by estimating the solvent-accessible surface area (*SASA*).

### 3.7. Weak Interaction Analysis

Weak interaction analysis refers to various forms of interactions such as electrostatic, hydrogen bonding, spatial repulsion, and van der Waals forces that are significantly weaker in strength than normal chemical bonds and can be used to discover non-covalent interactions between ligands and proteins [38]. In this paper, the averaged approximate density gradient (aRDG) method (also known as aNCI method) was used to graphically study the weak interactions. aRDG is valuable for studying protein−ligand interactions to visualize the averaged interactions in dynamic processes [38]. Therefore, for the study, the protein−ligand complex was first subjected to NPT simulation to bring the system to a fully equilibrated state. Subsequently, the ligand position was frozen and MD simulations with a duration of 1 ns were performed, and 1001 frames of trajectory simulations were extracted. Meanwhile, the aRDG of the protein−ligand complex was analyzed by Multiwfn software [39], and the box was defined by expanding 3 Å around the ligand and setting the grid point spacing to 0.15 [40]. The representation and color of the aRDG were displayed in the VMD software [41].

aRDG is calculated as follows [42]:(6)$$\mathrm{aRDG}\left(r\right)=\frac{1}{2{\left(3\pi^{2}\right)}^{1/3}}\frac{\left\lfloor \bar{\nabla \rho \left(r\right)}\right\rfloor }{{\bar{\rho \left(r\right)}}^{4/3}}$$

## 4. Conclusions

JAK kinases play an important role in the regulation of the immune system and are excellent targets for the treatment of autoimmune diseases. The main objective of this study was to redesign the new parent core to occupy a hydrophobic pocket and add a reversible covalent warhead, resulting in the novel, highly selective JAK3 inhibitors. The structure of the new parent core was combined with the characteristic structure of the lead compound PubChem CID 137321159, while a backbone growth approach was used to obtain an inhibitor centered on 1,7−dihydro−dipyrrolo [2,3−b:3′,2′−e] pyridine, using multiple docking approaches as well as ADMET predictions to narrow the list from 1468 down to six potential lead molecules. In addition, the binding patterns and interactions in docking studies were further evaluated by molecular dynamics simulations to understand their hydrogen-bonding patterns, binding free energies, and weak interactions. Finally, inhibitor 8 was found to enhance the interactions between the ligand and key amino acids of JAK3. Thus, the stability of the 5LWN−8 complex system increases, and this inhibitor is likely to have a high potential to inhibit the protein kinase from the theoretical view. Since this study was based only on multiple computational tools and simulation studies, these drugs need to be further investigated by in vitro and in vivo studies to confirm their activity against JAK3.

## Figures and Tables

**Figure 1 ijms-23-05437-f001:**
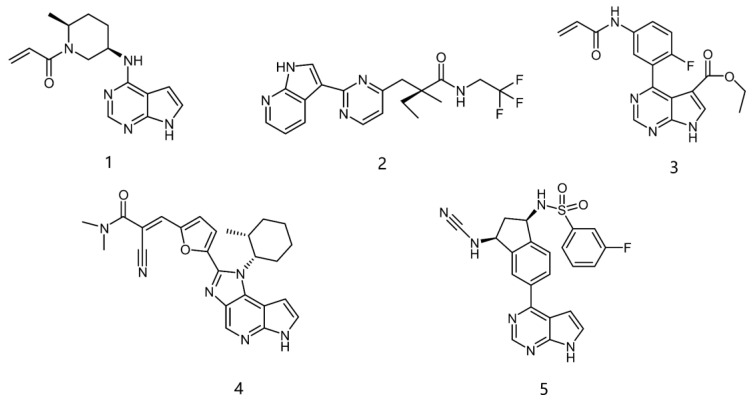
PF-06651600 [17] (**1**), a JAK3-specific inhibitor in clinical studies and recently reported JAK3 inhibitors [18] (**2**–**4**), PubChem CID 137321159 (**5**).

**Figure 2 ijms-23-05437-f002:**
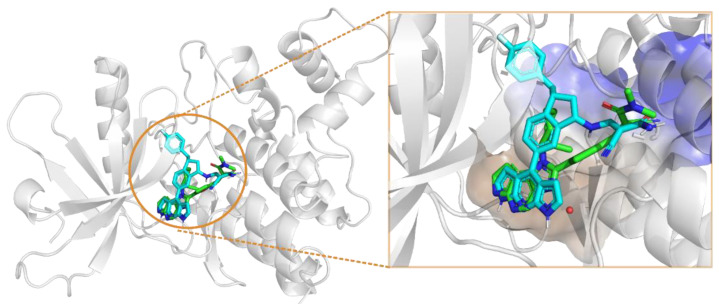
Stacking pattern of inhibitor 8 with eutectic ligand (inhibitor 4) in the JAK3 activity pocket, with inhibitor 8 in blue, inhibitor 4 in green, and water molecules in the solvent-exposed region in red.

**Figure 3 ijms-23-05437-f003:**
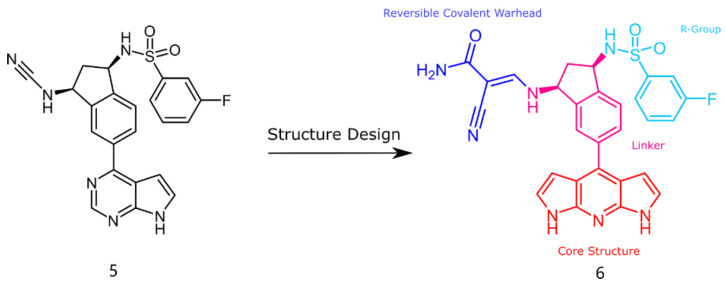
Inhibitor 6 was obtained by replacing the reversible covalent warhead and redesigning the core structure.

**Figure 4 ijms-23-05437-f004:**
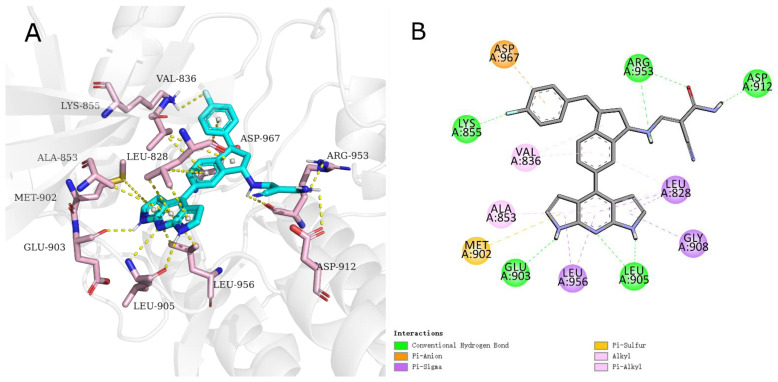
(**A**) Stereo view of the interaction of inhibitor 8 with residues near the active pocket of JAK3; (**B**) 2D map of the interaction of inhibitor 8 with key amino acids in JAK3.

**Figure 5 ijms-23-05437-f005:**
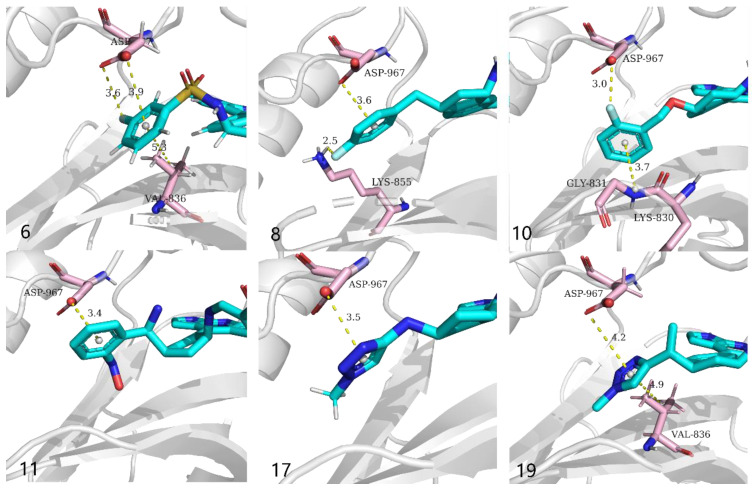
Three-dimensional map of the binding pattern of the R group of inhibitors 6, 8, 10, 11, 17, 19 to the active pocket of JAK3.

**Figure 6 ijms-23-05437-f006:**
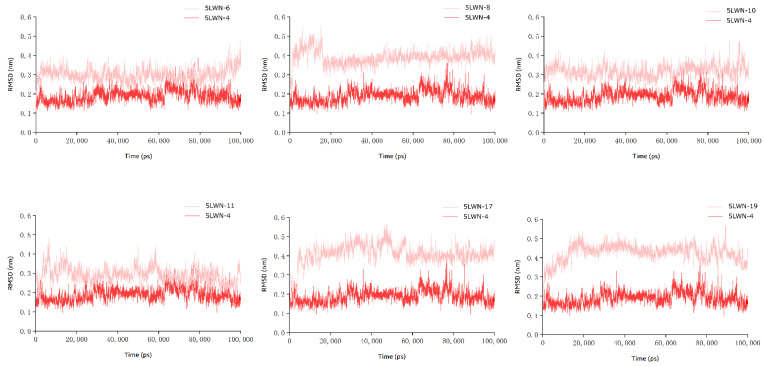
Comparison of RMSD of six inhibitors with eutectic ligands (79R) during molecular dynamics simulations.

**Figure 7 ijms-23-05437-f007:**
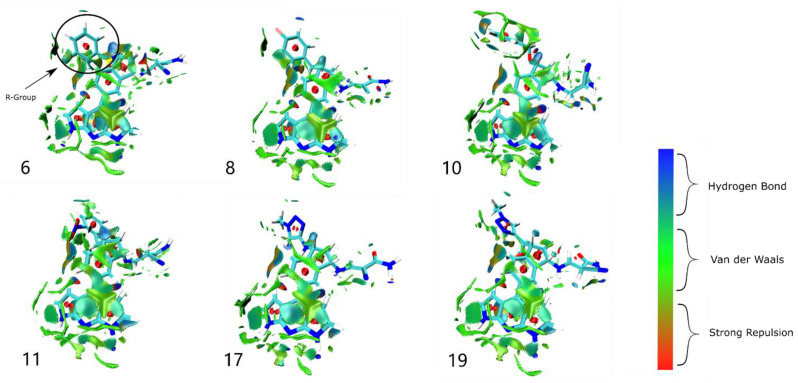
Analysis of the weak interaction of 1,7−dihydro−dipyrrolo [2,3−b:3′,2′−e] pyridine inhibitors with the JAK3 complex. Inhibitor numbers are represented by the numbers here.

**Figure 8 ijms-23-05437-f008:**
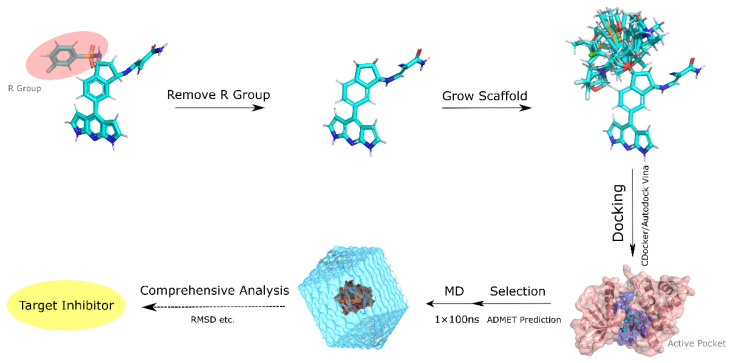
The flow chart describes the study to generate 1,7−dihydro−dipyrrolo [2,3−b:3′,2′−e] pyridine as the core inhibitor and screen the targeted inhibitor workflow.

**Table 1 ijms-23-05437-t001:** Structures of the top 14 inhibitor R-groups and the docking scoring and binding mode of the inhibitor to the receptor (PDB ID:5LWN).

Inhibitors	R	Affinity(kcal/mol)	Estimated Ki(nm)	H-Bond Interaction	Hydrophobic Interaction	Halogen Interaction
6	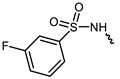	−10.1	39.51	GLU903, LEU905, ARG911, ARG953	LEU828, VAL836, ALA853, MET902, LEU956	ASP967
7	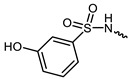	−10.8	12.12	GLU903, LEU905, AGR953, ASN954	LEU828, ALA853, VAL884, VAL836, MET902,LEU956, ALA966, ASP967	-
8	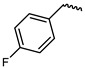	−11.8	2.24	LYS855, GLU903, LEU905, ASP912, AGR953	LEY828, VAL836, ALA853, LEU956, MET902, ASP967	ASP967
9	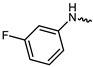	−11.1	7.31	LYS855, GLU871, GLU903, LEU905, ASP967	LEU828, VAL836, ALA853, MET902, LEU956	-
10	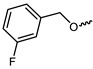	−11.0	8.65	GLU903, LEU905, ASP912, AGR953	LEU828, LYS830, VAL836, ALA853, MET902, LEU956	ASP967
11	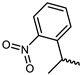	−11.1	7.31	GLU903, LEU905, AGR911, AGR953	LEU828, VAL836, ALA853, MET902, LEU956, ASP967	-
12	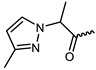	−11.0	8.65	LEU905, GLU903, AGR911, AGR953	LEU828, VAL836, ALA853, MET902, LEU956, GLY908, ASP967	-
13	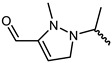	−11.2	6.17	LYS855, GLU903, LEU905, AGR911, ASP912, AGR953	LEU828, VAL836, ALA853, MET902, LEU956	-
14	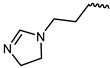	−10.8	12.12	LYS855, GLU903, LEU905, AGR911, ASP912, AGR953,	LEU828, VAL836, ALA853, MET902, GLY908, LEU956	-
15	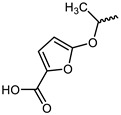	−10.7	14.35	LYS855, GLU903, LEU905, CYS909, AGR911, AGR953	LEU828, LYS830, VAL836, ALA853, VAL884,MET902, LEU956, ALA966	-
16	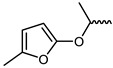	−11.8	2.24	LEU905, GLU903, AGR953, ASP912	LEU828, VAL836, ALA853, MET902, LEU956, ASP967	-
17	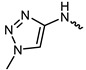	−11.3	5.21	GLU903, LEU905, ASP912, AGR953	LEU828, VAL836, ALA853, MET902, LEU956, ASP967	-
18	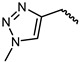	−11.3	5.21	GLU903, LEU905, ASP91, 2AGR953	LEU828, VAL836, ALA853, LYS855, MET902, LEU956, ASP967	-
19	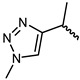	−11.6	3.14	GLU903, LEU905, ASP912, AGR953	LEU828, VAL836, ALA853, MET902, LEU956, ASP967	-

**Table 2 ijms-23-05437-t002:** The prediction of ADMET with 1,7-dihydro-dipyrrolo [2,3-b:3′,2′-e] pyridine as the core inhibitor.

Inhibitors	Lipinski Rules ^a^	HIA ^b^	PPB ^c^	BBB Permeant	CYP2D6 Inhibitor	hERG Blockers ^d^	Carcinogenicity ^e^
6	Accepted	0.012	95.78%	No	No	0.158	0.052
8	Accepted	0.005	97.90%	No	No	0.175	0.055
10	Rejected	0.007	96.23%	No	No	0.27	0.039
11	Rejected	0.009	98.23%	No	No	0.209	0.091
17	Accepted	0.014	89.07%	No	No	0.201	0.169
19	Accepted	0.006	93.07%	No	No	0.323	0.179

^a^: MW ≤ 500, logP ≤ 5, Hacc ≤ 10, Hdon ≤ 5; ^b^: 0–0.3: excellent, 0.3–0.7: medium, 0.7–1.0: poor; ^c^: ≤ 90%: excellent, otherwise: poor, ^d^: 0–0.3: excellent, 0.3–0.7: medium, 0.7–1.0: poor; ^e^: 0–0.3: excellent, 0.3–0.7: medium, 0.7–1.0: poor.

**Table 3 ijms-23-05437-t003:** Binding energies of 1,7-dihydro-dipyrrolo [2,3-b:3′,2′-e] pyridine-like inhibitors with JAKs and contributions to them.

Inhibitors	SASA Energy (kJ/mol)	Polar Solvation Energy (kJ/mol)	Electrostatic Energy (kJ/mol)	van der Waal Energy (kJ/mol)	Binding Free Energy (kJ/mol)
4	−21.532 +/− 1.219	192.312 +/− 18.531	−34.305 +/− 11.398	−202.871 +/− 14.174	−66.395 +/− 14.892
6	−22.127 +/− 1.023	248.181 +/− 22.567	−73.232 +/− 13.944	−204.047 +/− 13.286	−51.225 +/− 16.394
8	−20.542 +/− 0.875	202.732 +/− 11.996	−54.868 +/− 8.139	−197.608 +/− 12.352	−70.286 +/− 11.390
10	−20.840 +/− 1.026	248.490 +/− 23.490	−82.609 +/− 12.429	−186.017 +/− 13.120	−40.975 +/− 17.830
11	−21.146+/− 1.106	128.534 +/− 60.574	23.361 +/− 31.773	−195.273 +/− 14.230	−64.523 +/− 30.463
17	−20.403 +/− 0.974	244.416 +/− 19.635	−64.784 +/− 10.452	−202.051 +/− 12.772	−42.822 +/− 15.484
19	−20.165 +/− 1.000	245.281 +/− 23.425	−65.409 +/− 11.797	−199.462 +/− 13.097	−39.754 +/− 18.304

## Data Availability

All data are listed in tables or presented in figures in the main text or Appendix A.

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
