# Peer review of "Design of Rational JAK3 Inhibitors Based on the Parent Core Structure of 1,7-Dihydro-Dipyrrolo [2,3-b:3′,2′-e] Pyridine"

_ijms, 2022, doi:10.3390/ijms23105437_

Round 1

Reviewer 1 Report

The authors presented Design of rational JAK3 inhibitors based on the parent core 2

structure of 1,7-dihydro-dipyrrolo[2,3-b:3',2'-e] pyridine.  These is interesting data and the article in a good production and can be accepted after minor revision but after considering the following points 

My comments

1- Aim of the work should be stated clearly in intoduction

2- It is more convenient to make a comparison with other related studies in literature

3- The authors must revise language of the manuscript before publication and the whole article should be adjusted based on journal style.

Author Response

Dear reviewer:

We are grateful for the reviewers’ effort in reviewing our manuscript entitled " Design of rational JAK3 inhibitors based on the parent core structure of 1,7-dihydro-dipyrrolo[2,3-b:3',2'-e] pyridine" (ID:ijms-1730277). Those comments are very helpful for revising and improving this manuscript. We have considered the comments carefully and made corrections which we hope meet with approval. The main corrections are in the manuscript and the responses to the reviewers’ comments are as follows.

Point 1- Aim of the work should be stated clearly in introduction

Response 1:

We have added “To this end, we have used PubChem CID 137321159 as the characteristic part of the lead inhibitor for assembling and redesigning the new parent nucleus structure to fill the hydrophobic pocket as well to deliver reversible covalent warheads into the expected sites.” in lines 76-79 to specifically state the aim of our study in the introduction.

Point 2- It is more convenient to make a comparison with other related studies in literature

Response 2:

Yes, it is correct that we should make a comparison with other related studies in the literature. Thus, in lines 63-76, we have added a corresponding review- “Thorarensen et al. reported PF-06651600, an inhibitor based on a 6,7-dihydro-5H-pyrrolo[2,3-d]pyrimidine structure with an acrylamide warhead that irreversibly binds to Cys909 in JAK3. However, the collective consequences of inhibiting JAK3 in a completely irreversible covalent manner remain unclear and are not beneficial for all purposes[13]. London et al. sidestepped this problem by designing reversible covalent JAK3 inhibitors using cyanoacrylate warheads[14]. Forster et al. built on this to design reversible covalent JAK3 inhibitors with a tricyclic structure that exhibited both high isomeric and dynamic group selectivity and strong cellular activity and selectivity[15]. In addition, structural optimization of AZD1480 by AstraZeneca revealed that filling the hydrophobic pocket around the hinge region significantly improved the inhibition of JAK2 by the compound[16], and thus this principle could be applied to selective JAK3 inhibitors. Unfortunately, however, none of the JAK3 parent nucleus structures reported in the literature can fully utilize the solvent-exposed region of the ATP-active pocket and fill the hydrophobic pocket.”.

Point 3- The authors must revise language of the manuscript before publication and the whole article should be adjusted based on journal style.

Response 3:

We have revised the language of the manuscript and adjusted the reference style based on the journal style.

Reviewer 2 Report

In general, the manuscript is well presented and includes all the data and methods required. One minor remark about presentation. The choice of colors in Fig.7 is not the best. It is very complicated to recognize  strength of interactions, since all of them have more\less the same magnitude.  If possible, please use a more sharp distribution of colors. If not, remain as is.

Author Response

Dear reviewer:

We are grateful for the reviewers’ effort in reviewing our manuscript entitled " Design of rational JAK3 inhibitors based on the parent core structure of 1,7-dihydro-dipyrrolo[2,3-b:3',2'-e] pyridine" (ID:ijms-1730277). We have studied the comments carefully and made corrections which we hope meet with approval. The main corrections are in the manuscript and the responses to the reviewers’ comments are as follows.

Point 1: In general, the manuscript is well presented and includes all the data and methods required. One minor remark about presentation. The choice of colors in Fig.7 is not the best. It is very complicated to recognize strength of interactions, since all of them have more\less the same magnitude.  If possible, please use a more sharp distribution of colors. If not, remain as is.

Response 1: We have carefully considered the suggestion of the reviewer in order to make some changes. As suggested by the reviewer, better colors should be used in Fig.7. The VMD software, however, limits the ability to change the color of the image. We first used Multiwfn software to analyze the weak protein-ligand interactions during kinetics, and then VMD software to graph the results. Although the program allows you to change the color scale range, this is likely to result in distorted images. As a result, we decided to keep it as is. A comparable image can be seen in the literature [1].

  1. Zhang, J.; Zhang, L.; Xu, Y.; Jiang, S.; Shao, Y. Deciphering the binding behavior of flavonoids to the cyclin dependent kinase 6/cyclin D complex. PLoS One 2018, 13, e0196651, doi:10.1371/journal.pone.0196651.
